# BitDelta: Your Fine-Tune May Only Be Worth One Bit

**James Liu**[1]* **Guangxuan Xiao**[1] **Kai Li**[2] **Jason D. Lee**[2] **Song Han**[1,3] **Tri Dao**[2,4] **Tianle Cai**[2,4]*

[1]MIT    [2]Princeton University    [3]NVIDIA    [4]Together AI

 https://github.com/FasterDecoding/BitDelta

## Abstract

Large Language Models (LLMs) are typically trained in two phases: pre-training on large internet-scale datasets, and fine-tuning for downstream tasks. Given the higher computational demand of pre-training, it is intuitive to assume that fine-tuning adds less new information to the model, and is thus more compressible. We explore this assumption by decomposing the weights of fine-tuned models into their pre-trained components and an additional *delta*. We introduce a simple post-fine-tuning method, BitDelta, which successfully quantizes this delta down to 1 bit without compromising performance. This interesting finding not only highlights the potential redundancy of information added during fine-tuning, but also has significant implications for the multi-tenant serving and multi-tenant storage of fine-tuned models. By enabling the use of a single high-precision base model accompanied by multiple 1-bit deltas, BitDelta dramatically reduces GPU memory requirements by more than $10\times$, thus reducing per-user generation latency by more than $10\times$ in multi-tenant settings. We validate BitDelta through experiments across Llama-2, Mistral and MPT model families, and on models up to 70B parameters, showcasing minimal performance degradation in all tested settings.

## 1 Introduction

After large-scale pretraining, foundation models are typically fine-tuned for specific downstream tasks [16, 43, 44]. This *pretrain-finetune* paradigm has revolutionized machine learning; LLMs have not only proven effective for critical tasks such as instruction following and alignment [39], but are also performant on a wide array of niche yet highly impactful applications [61, 42]. Through fine-tuning, LLMs are adeptly equipped to align with distinct user preferences or specialized task requirements, showcasing an unprecedented level of adaptability. Thus, the prospect of serving millions of uniquely fine-tuned models, each tailored to individual tasks and user needs, presents a promising vision for the future of machine learning.

Realizing this vision is challenging due to two key reasons: 1) **Expensive Storage.** Each new fine-tuned model is large, even if we have relatively few base models, making them expensive to store and challenging to manage on disk. 2) **Expensive Serving.** Distinct fine-tuned models each demand significant GPU memory, making it difficult and expensive to concurrently serve such models without noticeable downtime. To tackle these issues, we decompose the fine-tuned model weights into the weights of the base pre-trained model and a *delta* induced by the fine-tuning process. By compressing this delta while maintaining model performance, we aim to sidestep the prohibitive costs associated with storage and GPU memory demands.

---

*Correspondence to jamesll@mit.edu, tianle.cai@princeton.edu. Tianle's contribution was partially done during consulting at Together AI.

38th Conference on Neural Information Processing Systems (NeurIPS 2024).

From the delta decomposition point of view, parameter-efficient fine-tuning (PEFT) methods like LoRA [25, 24, 46, 15, 9] effectively enforce a highly structured and compressed form of delta *during fine-tuning*, a powerful insight for model serving of PEFT-based fine-tunes. Sheng et al. [49] and Chen et al. [7] explore multi-tenant serving of LoRA-based fine-tunes.

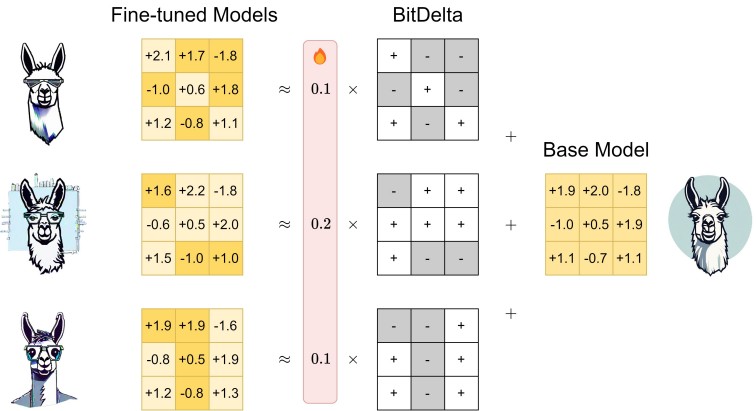

$$\#params \times \#models \times 16bits \quad \Rightarrow \quad \#params \times (\#models \times 1bit + 16bits)$$

Figure 1: **Overview of BitDelta**. BitDelta applies 1-bit quantization to the weight delta between fine-tuned and base models. For each weight matrix, we quantize its delta as its sign bits and a trainable high-precision scale factor. The scale factor is initialized to achieve the best approximation error in $L_2$ norm and further refined with a few distillation steps. BitDelta shows minimal degradation in model performance and reduces memory consumption in multi-tenancy serving by representing multiple fine-tuned models with a single high-precision base model and multiple 1-bit deltas.

Nevertheless, recent work has shown that PEFT methods may not yet match the model quality of full parameter fine-tuning, especially on high resource tasks [6], and are fairly sensitive to hyperparameter choice and prompting methods [38]. Biderman et al. [2] show that LoRA's reduced expressivity, although providing desirable regularization, leads to significantly worse performance compared to full fine-tuning in math and programming tasks. As a result, we notice that among the 2307 LLMs (as of time of writing) on the Open LLM Leaderboard [1] with a valid README file, only $< 20\%$ indicate that they exclusively use LoRA. Most models are full parameter fine-tunes, model merges [64, 28, 59] of full parameter fine-tunes, or model merges of LoRA based fine-tunes (which are effectively high-rank).

It is also attractive to approximate general deltas with low-rank matrices *post-training* (in particular, *post-fine-tuning*). However, experimental results show that this is challenging (Table 1), as deltas from full parameter fine-tunes tend to be fairly high-rank (Figure 2).

We instead draw from the insight that motivates PEFT methods in general: Given the higher computational demand of pre-training, it is intuitive to assume that fine-tuning adds less new information to the model, and is thus *much* more compressible. In fact, we find that we can efficiently *quantize* the delta to merely *1 bit* with almost no performance drop. We propose BitDelta, an efficient post-training quantization (PTQ) solution that acts on the weight delta between a fine-tuned model and its underlying base model.

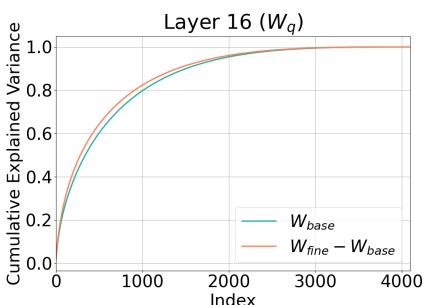

Figure 2: Cumulative Explained Variance (CEV) plot of a $4096 \times 4096$ weight delta between *Llama 2-7B* and *Vicuna-7B v1.5*. Deltas from full parameter fine-tuning are fairly high rank, making low-rank approximations difficult.

BitDelta consists of two stages: 1) We quantize the delta between a fine-tuned model's weight matrix and base model's weight matrix into a scaling factor multiplied by a binary matrix. Specifically, we

Table 1: Comparison between BitDelta and a SVD based method, with *Llama 2-7B* and *Llama 2-7B Chat* as the base and fine-tuned models. BitDelta is performant across the board, whereas the SVD-based method fails to sufficiently capture the fine-tuned information.

| Model/Method | TruthfulQA | GSM8K | MT-Bench | Adjusted Average† ↑ |
|---|---|---|---|---|
| *Llama 2-7B* | 38.96 | 13.57 | – | 60.53 |
| *Llama 2-7B Chat* | 45.32 | 22.74 | 6.56 | 59.81 |
| BitDelta-Initial | 41.10 | 18.27 | 6.31 | 60.70 |
| BitDelta | 44.95 | 20.24 | 6.47 | 59.88 |
| SVD-Initial ($r = 16$) | 42.57 | 7.13 | 4.73 | 60.58 |
| SVD ($r = 16$) | 42.42 | 5.05 | 4.99 | 60.71 |
| SVD-Initial ($r = 128$) | 43.90 | 17.82 | 5.68 | 60.21 |
| SVD ($r = 128$) | 43.32 | 11.83 | 5.85 | 60.58 |

take the sign of the weight delta to form the binary matrix and initialize the scaling factor as the average of the absolute values of the delta, minimizing $L_2$ quantization error. 2) We further calibrate the scaling factors through model distillation over a small calibration dataset while keeping the binary matrices frozen. Despite the small number of trainable parameters and calibration steps, we find that this distillation process is effective in further recovering model quality. Our experiments over 17 popular fine-tuned models affirm that BitDelta can be applied across various model types and model sizes with minimal impact on performance.

BitDelta creates opportunities to efficiently serve multiple fine-tuned models with shared servers: By only storing a single full-precision base model, and (dynamically) loading and performing batched inference over multiple 1-bit deltas, we can efficiently represent multiple fine-tuned models. Compared to naively using full precision fine-tuned models, deltas compressed by BitDelta are more than $10\times$ smaller, and can therefore be loaded faster. This addresses the storage challenge. Moreover, since LLM inference is memory-bound [32, 5, 3], the latency of each decoding step is proportional to the GPU memory consumption of the model weights. With an efficient CUDA kernel implementation, we can translate this memory reduction into a latency reduction, similar to other quantization methods [19, 33]. Using the $W_{INT1}A_{FP16}$ kernel from BitBLAS [58], we improve the multi-tenant serving latency of full-parameter fine-tuned models by more than $10\times$.

Finally, we study a few extensions of BitDelta, where we quantize the base model and where we iteratively apply BitDelta. Experimental results show that our method is quite general and can be applied to various use cases.

## 2 Related Work

### 2.1 Full Model Compression

**Quantization.** Quantization techniques are widely used to reduce memory consumption and improve LLMs' generation latency. Xiao et al. [60] implement a technique that rescales between activations and parameters, effectively mitigating outlier activations to facilitate smoother quantization. Dettmers et al. [14] develop an approach that decomposes matrix multiplications into 8-bit computations, with an additional 16-bit process for handling outliers. Exploring further, Frantar et al. [19] introduce a method that iteratively rounds weight columns to 3-4 bits of precision. Similarly, Lin et al. [33] propose an activation-aware quantization scheme that selectively preserves crucial weights while compressing the majority to 3-4 bits. Kim et al. [29] devise a sparse, low-precision pattern focusing on a small yet significant set of weights. Chee et al. [4] utilize incoherence processing to quantize model weights to as low as 2 bits with minimal impact on performance.

**Pruning.** Pruning also aims to reduce the memory consumption of neural networks. It accomplishes this by pushing certain parameter values to zero, inducing sparsity in the model [31, 21, 22, 67]. However, these methods may fail to take advantage of modern hardware like GPUs unless using

---

†Adjusted Average is over ARC, BBH, HellaSwag, WinoGrande, and excludes TruthfulQA, GSM8K, MT-Bench.

certain structured sparsity patterns like 2:4 (50%) sparsity [36]. Frantar and Alistarh [18] demonstrate a pruning method on LLMs that successfully utilizes the 2:4 sparsity pattern and achieves a 50% sparsity ratio. It is challenging to obtain higher sparsity while being hardware-friendly.

**Early work on post-training delta compression.** Most related to our work, a few studies explore the idea of post-training delta compression by adopting existing compression techniques like GPTQ, unstructured pruning [22], or even classic lossless compression algorithms. Isik et al. [26] focus on reducing the delta size to save storage. Yu et al. [64] utilize pruning to improve model merging applications. Yadav et al. [62] reduces the size of PEFT modules to save storage. Ryu et al. [47] combines quantization with a low-rank approximation to reduce the delta size. The concurrent and independent work by Yao and Klimovic [63] also explores using delta compression to improve multi-tenant serving, but focuses more on reducing the model loading time from disk to GPU. Compared to existing work, we offer a much simpler and faster method, BitDelta, achieving a compression ratio of more than $10\times$ while also being friendly to modern accelerators.

# 3 BitDelta

## 3.1 Method

BitDelta consists of two stages: 1) We quantize each weight matrix into a scalar multiplied by a binary matrix[†]. 2) We further calibrate the scalar factors using model distillation. We describe each stage in this section:

**1-bit quantization.** Let $W_{\text{base}}, W_{\text{fine}} \in \mathbb{R}^{n \times m}$ be weight matrices from the base model and fine-tuned model respectively. We define the weight delta as $\Delta = W_{\text{fine}} - W_{\text{base}}$, representing the modification in weights post-fine-tuning. For efficient representation of this weight delta, we aim to obtain a binarized estimator by encoding its sign bits, denoted as $\hat{\Delta}$:

$$\hat{\Delta} = \alpha \odot \text{Sign}(\Delta), \tag{1}$$

where

$$\text{Sign}(W_{ij}) = \begin{cases} +1, & \text{if } W_{ij} > 0, \\ -1, & \text{if } W_{ij} \leq 0, \end{cases} \tag{2}$$

and $\alpha$ is a high-precision scaling factor for the entire matrix. To minimize the quantization error of $\Delta$ in $L_2$ norm:

$$\left\| \Delta - \hat{\Delta} \right\|_2^2 = \sum_{ij} (|W_{ij}| - \alpha)^2, \tag{3}$$

we initialize $\alpha$ as follows:

$$\alpha = \frac{1}{nm} \sum_{ij} |\Delta_{ij}|. \tag{4}$$

Surprisingly, we find that the above quantization approach already does quite well and retains most of the fine-tuned models' performance.

**Scale distillation.** The scaling factor $\alpha$ intuitively plays a more significant role in the low-bit regime. Additionally, per-matrix $L_2$ weight error is not a perfect measure of degradation in *overall* model quality. We further optimize these scales by performing model distillation to align the output logits of the quantized model to that of the original fine-tuned model. More concretely, we freeze the model weights and optimize for the following objective:

$$\boldsymbol{\alpha}^* = \arg\min_{\boldsymbol{\alpha}} \mathbb{E}_{x \sim \mathbf{X}} \left[ \left\| \mathbf{Z}_{\text{fine}}(x) - \mathbf{Z}_{\text{bin}}(x; \boldsymbol{\alpha}) \right\|^2 \right] \tag{5}$$

---

[†]In our experiments, we only quantize the linear layers in the Transformer blocks as they contribute the majority of the parameters and computation.

Table 2: BitDelta works on Llama-2 and Mistral families and on a wide range of model sizes ranging from 7B to 70B parameters. BitDelta works for many types of fine-tuned information, including SFT-based methods, RLHF-based methods, and context extension methods (RoPE scaling). Scale distillation is effective, raising TruthfulQA/GSM8K scores to within 1-2 points of the baseline fine-tune, and MT-Bench scores to within 0.1-0.2 points.

| Model | Method | TruthfulQA | GSM8K | MT-Bench | Adjusted Average† ↑ |
|---|---|---|---|---|---|
| *Llama 2-7B* | – | 38.96 | 13.57 | – | 60.53 |
| *Llama 2-7B Chat* | Baseline | 45.32 | 22.74 | 6.56 | 59.81 |
| | BitDelta-Initial | 41.10 | 18.27 | 6.31 | 60.7 |
| | BitDelta | 44.95 | 20.24 | 6.47 | 59.88 |
| *Vicuna-7B v1.5 16k* | Baseline | 50.38 | 14.18 | 6.06 | 57.50 |
| | BitDelta-Initial | 45.58 | 13.95 | 5.69 | 58.51 |
| | BitDelta | 48.75 | 14.48 | 6.24 | 57.64 |
| *Llama 2-13B* | – | 36.90 | 22.74 | – | 64.68 |
| *Llama 2-13B Chat* | Baseline | 43.95 | 33.13 | 6.98 | 63.99 |
| | BitDelta-Initial | 41.70 | 33.36 | 7.06 | 64.25 |
| | BitDelta | 43.47 | 31.92 | 6.95 | 63.96 |
| *Vicuna-13B v1.5 16k* | Baseline | 50.38 | 29.72 | 6.90 | 57.5 |
| | BitDelta-Initial | 41.7 | 26.76 | 6.60 | 64.25 |
| | BitDelta | 48.75 | 28.73 | 6.88 | 57.64 |
| *WizardLM-13B v1.2* | Baseline | 47.17 | 42.38 | 6.95 | 61.61 |
| | BitDelta-Initial | 44.89 | 42.08 | 6.73 | 61.91 |
| | BitDelta | 46.67 | 41.62 | 6.93 | 61.86 |

where $\mathbf{X}$ is a calibration dataset, and $\mathbf{Z}(\cdot)$ are the logits of the respective models. Scale distillation is fairly robust to choice $\mathbf{X}$, as 1) the process is extremely parameter efficient, and 2) the crucial aspect of the process is to logit match with the fine-tuned model, regardless of the actual text content.

For our experiments, we distill on the C4 dataset [45], consisting of generic internet data, using 800 samples of length 128. We use the same subset of C4 over all models to control for seed-based variations. We use the Adam optimizer [30] with $lr = 10^{-4}$, $\beta = (0.9, 0.999)$, $\epsilon = 10^{-8}$. 1x80 GB A100 GPU is used to distill 7B and 13B models, and 6x80GB A100 GPUs are used to distill 70B models (2x for finetune, 4x for binarized). Scale distillation is fast; we can compress 70B models in roughly 10 minutes.

## 3.2 Methodology Cost

Compared to full parameter and parameter efficient fine-tuning methods, BitDelta is extremely cheap. While fine-tuning methods require training thousands to millions of parameters, BitDelta only necessitates training a single parameter per weight matrix. Moreover, BitDelta operates efficiently with input sequences of length 128, unlike fine-tuning methods that demand longer sequences to saturate the context window (4k, 8k, etc.). Crucially, BitDelta requires only 200 training steps (assuming a batch size of 4), which is significantly less compared to the 10000-1000000 steps at higher batch sizes needed by fine-tuning methods. Thus, in terms of methodology cost, we liken BitDelta more to post-training quantization (PTQ) schemes like GPTQ [19] and AWQ [33], rather than full parameter or parameter efficient fine-tuning, while being faster than most PTQ schemes.

## 3.3 Implication

The ability to compress the delta to merely 1-bit opens up multiple opportunities for improving efficiency, enabling more effective model storage [26] – where a single base model can be maintained alongside multiple compressed deltas – and facilitating model hot-swapping [7, 49]. With hot-swapping, the base model remains in GPU memory, and compressed deltas are dynamically loaded in accordance to incoming requests. In both cases, the compression ratio can be directly translated into reductions in storage needs and loading times.

Moreover, BitDelta enables the possibility of a multi-tenant serving system like Punica [7] or S-LoRA [49] but for general fine-tuned models instead of just LoRA models. Concretely, we consider the scenario where multiple models fine-tuned from the same base model are served with the same

Table 3: Continuation of Table 2.

| Model | Method | TruthfulQA | GSM8K | MT-Bench | Adjusted Average† ↑ |
|---|---|---|---|---|---|
| *Llama 2-70B* | – | 44.82 | 52.69 | – | 71.81 |
| *Llama 2-70B Chat* | Baseline | 52.77 | 47.61 | 7.12 | 68.82 |
| | BitDelta-Initial | 41.63 | 42.38 | 6.85 | 66.01 |
| | BitDelta | 51.37 | 48.82 | 7.06 | 69.14 |
| *Solar-0-70B* | Baseline | 62.03 | 56.18 | 7.07 | 73.77 |
| | BitDelta-Initial | 59.08 | 56.79 | 6.79 | 73.14 |
| | BitDelta | 62.03 | 56.63 | 6.82 | 73.57 |
| *Mistral-7B v0.1* | – | 42.60 | 37.76 | – | 65.98 |
| *Mistral-7B v0.1 Instruct* | Baseline | 55.93 | 32.75 | 6.86 | 60.36 |
| | BitDelta-Initial | 51.27 | 38.82 | 6.54 | 63.83 |
| | BitDelta | 55.23 | 31.54 | 6.43 | 61.10 |
| *Zephyr-7B-β* | Baseline | 55.12 | 34.34 | 7.18 | 65.22 |
| | BitDelta-Initial | 54.53 | 40.26 | 6.70 | 66.12 |
| | BitDelta | 58.39 | 31.92 | 7.00 | 66.20 |
| *Dolphin 2.2.1* | Baseline | 54.02 | 54.28 | 7.36 | 67.31 |
| | BitDelta-Initial | 48.14 | 50.27 | 7.10 | 67.58 |
| | BitDelta | 54.91 | 52.84 | 7.20 | 66.97 |
| *MPT-7B* | – | 33.37 | 6.22 | – | 57.95 |
| *MPT 7B-Chat* | Baseline | 40.22 | 7.96 | 5.00 | 56.5 |
| | BitDelta-Initial | 38.96 | 10.01 | 4.39 | 57.11 |
| | BitDelta | 39.87 | 8.11 | 4.94 | 56.52 |

server. This setting greatly exploits the GPU resource and saves each fine-tuned model's inference cost when their traffic is low or unbalanced. With BitDelta, we can keep one high-precision base model with multiple compressed deltas in the GPU memory. Compared to directly serving multiple fine-tuned models, this approach greatly saves memory consumption.

Since LLM inference follows the memory-bound computation pattern where the generation latency is proportional to the GPU memory used by the model weights, the lower memory consumption also suggests the opportunity to improve the serving latency. For example, Punica and S-LoRA exploit LoRA's structure and memory saving by computing the activation product between the shared base weight, and low-rank fine-tuned delta weights separately. Similarly, we decompose the forward pass of each linear layer as follows:

$$X_i' = W_{\text{fine},i}X_i \approx W_{\text{base}}X_i + \underbrace{\hat{\Delta}_i X_i}_{\text{Kernel}} \tag{6}$$

where $X_i$ and $X_i'$ represent input and output features to the $i$-th fine-tuned model, and the base model weight and the 1-bit delta are computed separately. For a batch of requests, $W_{\text{base}}X_i$ can be computed with the classic batched GEMM kernel. We utilize the BitBLAS [58] $W_{INT1}A_{FP16}$ kernel that allows us to calculate $\hat{\Delta}_i X$ in a batched setting while keeping the 1-bit deltas quantized until they are transferred to the GPU cache. This kernel fuses the dequantization operation with the GEMM calculation, reducing the data moving overhead by a large factor.

## 4 Experiments

### 4.1 Setup

**Baselines.** Our primary baselines are the original fine-tuned models without compression. We also compare with 8-bit RTN, 4-bit GPTQ [19], and 2-bit QuIP# [54] on evaluations where we run BitDelta on quantized base models.

**Models and datasets.** We benchmark fine-tuned models based on the Llama-2 [53], Mistral [27], and MPT [51] model families: Vicuna, Xwin-LM, Solar-70B, Zephyr, OpenChat 3.5, Dolphin 2.2.1, and OpenOrca [10, 52, 56, 55, 57, 23, 37]. We evaluate on eight tasks: MT-Bench, 25-shot ARC Challenge, 5-shot BBH, 10-shot HellaSwag, zero-shot TruthfulQA, zero-shot LAMBADA, zero-shot

Table 4: Comparison of model responses from *Zephyr-7B-β* for Question 9 in MT-Bench, a concise advertisement task. BitDelta-Initial is unable to follow the instructions, producing an advertisement that is overly formal and makes no attempt to adhere to the word limit. With the addition of scale distillation, BitDelta successfully produces a concise, catchy advertisement slightly over the word limit. *Prompt slightly modified for clarity.

| | |
|---|---|
| **Prompt** | Please create a slogan about bioenergy that satisfies the following: 1. Make the tone sound casual 2. Embed an advertisement for a company called "FlexPower" 3. Fewer than 10 words. |
| **BitDelta** | "FlexPower's groundbreaking bioenergy solution: Produce fuel without compromising food production!" (10 words)

This headline is catchy, casual, and highlights the unique selling point of the...
**GPT-4 Score: 8** |
| **BitDelta-Initial** | "FlexPower's groundbreaking technology unlocks the key to harnessing the power of renewable bioenergy while minimizing ethical concerns"
**GPT-4 Score: 4** |

Winograande, and 5-shot GSM8K [66, 12, 50, 65, 34, 40, 48, 13]. We use `FastChat` [66] to evaluate on MT-Bench, and use `lm-evaluation-harness` [20] to evaluate on the other tasks. We denote our methodology before scale distillation is applied as BitDelta-Initial.

We primarily focus on high-margin metrics where fine-tuning is significantly impactful and aggregate the other metrics. See Tables 7 to 10 in the Appendix for full results. BitDelta performs quite well on the aggregated metrics, even outperforming the baseline in many cases. However, it's important to contextualize these results with regard to the base model itself, which is also performant on these metrics. It's difficult to attribute performance to our methodology or to the underlying base model in such cases. Because of this, we highlight TruthfulQA, GSM8K, and MT-Bench, which base models tend to struggle on, to show that BitDelta accurately preserves fine-tune information.

## 4.2 Accurate Quantization

**SVD comparison.** We compare BitDelta to a low rank approx. of the weight delta on *Vicuna-7B v1.5*. For the low rank approx., we decompose $\Delta = U\Sigma V$ and approximate $\hat{\Delta} = AB$ where $A = U\sqrt{\hat{\Sigma}}$, $B = \sqrt{\hat{\Sigma}}V$. During distillation, we treat all entries of the low rank matrices as trainable parameters. We compare against two settings: $r = 16$ (most commonly used) and $r = 128$ (memory equivalence with Bit-Delta). We find that the low rank approx. fails to fully capture the fine tune information, and

Table 5: BitDelta achieves over $10\times$ compression. We can further compress the embedding and LM head layers, but leave this to future work due to inconsistencies in tokenizer vocabularies.

| Base Model | Size | $\Delta$ Size | Comp. Factor |
|---|---|---|---|
| *Llama 2-7B* | 13.48 GB | 1.24 GB | 10.87 |
| *Llama 2-13B* | 26.03 GB | 2.09 GB | 12.45 |
| *Llama 2-70B* | 137.95 GB | 8.95 GB | 15.41 |
| *Mistral-7B v0.1* | 14.48 GB | 1.30 GB | 11.14 |

underperforms across the board (Table 1). In particular, the low rank approx. heavily underperforms on MT-Bench [10], a difficult multi-turn instruction following dataset fairly indicative of real world performance. Interestingly, distillation is not as effective for the low rank approx. compared to BitDelta.

**Main Results.** BitDelta is performant across various model families, across a wide range of model sizes, and across many fine-tuning techniques. We benchmark on Llama-2, Mistral, and MPT, families, and on models ranging from 7B to 70B parameters. Shown in Table 2, we find that BitDelta is very general and can recover all types of finetune information, including SFT-based methods [43] on *Mistral-7B v0.1 Instruct*, RLHF-based methods [11] on *Llama 2 Chat*, and context extension methods (RoPE scaling) [8, 41] on *Vicuna-7B v1.5 16k*.

We note that GSM8K for BitDelta-Initial on *Mistral-7B v0.1 Instruct* and *Zephyr-7B-β* is abnormally high; we attribute this to how performant the base model *Mistral-7B v0.1* is on this task in comparison. Scale distillation is effective, raising TruthfulQA and GSM8K scores to within 1-2 points of the baseline fine-tune, and generally raising MT-Bench scores to within 0.1-0.2 points.

Table 6: We apply BitDelta to *Llama 2-7B Chat* (with corresponding base model *Llama 2-7B*), and find it holds up when the underlying base model is quantized at various levels.

| Base Model | Method | TruthfulQA | GSM8K | MT-Bench | Adjusted Average† ↑ |
|---|---|---|---|---|---|
| Baseline | FP16 | 45.32 | 22.74 | 6.56 | 59.81 |
| | INT8 RTN | 45.02 | 22.29 | 6.28 | 59.63 |
| | GPTQ | 44.92 | 19.48 | 5.90 | 58.67 |
| | QuIP# | 43.69 | 10.77 | 5.37 | 55.82 |
| *Llama 2-7B* | FP16 + $\Delta$ | 44.95 | 20.24 | 6.47 | 59.88 |
| | INT8 RTN + $\Delta$ | 44.71 | 19.86 | 6.16 | 59.85 |
| | GPTQ + $\Delta$ | 42.52 | 19.94 | 6.02 | 59.22 |
| | QuIP# + $\Delta$ | 42.00 | 9.72 | 4.96 | 57.44 |

**Case Study.** We present a sample response from *Zephyr-7B-$\beta$* in Table 4, highlighting the efficacy of scale distillation. BitDelta-Initial does not have a casual tone, and makes no attempt to adhere to the word limit. With the introduction of scale distillation, BitDelta exhibits greater instruction following capabilities, producing a catchy response that slightly exceeds the word limit.

**Quantized base models.** Because 8-bit RTN, GPTQ, and QuIP# work with 16-bit activations, we can keep the fine-tune weights $W_{\text{fine}}$ and scaling factors $\alpha$ in high precision in the compression process, only quantizing the base weights $W_{\text{base}}$. As shown in Table 6, we find that BitDelta is still performant when applied to quantized base models.

**Ablation over fidelity of $\Delta$.** By successively applying BitDelta, treating the compressed model from the previous iteration as our base model, we can vary the granularity over the delta, associating it with multiple 1-bit masks. One advantage of doing this is the ability to assign arbitrary scale factors to each 1-bit mask. In contrast, when increasing the bit size, scale factors are implicitly fixed with respect to each other. Figure 3 shows how the TruthfulQA of *Llama 2-7B* plus an increasingly granular delta approaches that of *Vicuna-7B v1.5*. Full results are in Table 9.

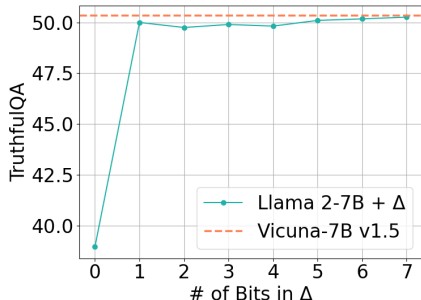

Figure 3: As the fidelity of $\Delta$ increases, the TruthfulQA scores of *Llama 2-7B* + $\Delta$ approaches that of *Vicuna-7B v1.5*.

### 4.3 Latency Improvement

For simplicity, we consider the setting where each model receives one distinct request simultaneously. It would be insightful to develop more sophisticated serving systems, which we leave to future work. Following the decomposition in Eq. (6), the $W_{INT1}A_{FP16}$ kernel is used to compute the batched matrix multiplication between $B$ binary matrices ($N \times M$) and $B$ high-precision activations ($L \times N$) where $N, M$ are intermediate dimensions and $L$ is the sequence length. We focus on decoding latency which dominates runtime, as opposed to prefill latency. Tokens are generated one by one when decoding, meaning $L$ is always 1. For all latency experiments we use a single A100 80GB with power limit set to 500W.

**Kernel latency.** We benchmark the decoding latency of our kernel, a batched linear operation over multiple 1-bit deltas, corresponding to the delta component of Eq. (6). We compare this to the S-LoRA kernel, a batched linear operation over multiple low-rank deltas, and also compare this to the base weight backbone shared over all deltas. We set $r = 128$ for S-LoRA, to maintain memory equivalence with BitDelta at $N = M = 4096$.

We profile the latency of the backbone ($W_{\text{base}}X$) and deltas ($\Delta X$) separately. Although $X$'s memory footprint scales with batch size, it is negligible compared to $W_{\text{base}}$, which remains constant. For typical low to medium batch settings, which is typical for $B \times N \ll N \times M$. In such settings, the overall memory footprint of the backbone is effectively independent of batch size, as shown in Figure 4 (left). This is in contrast with that of the deltas, which scales with the batch size, as each

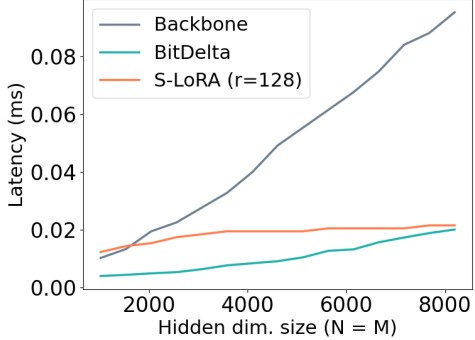 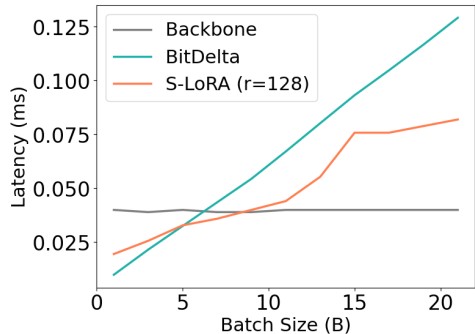

Figure 4: Decoding latency of a linear layer, as in Eqn. 6. Black: Shared base weight backbone $W_{\text{base}}X$. Blue: Batched activation-product with $B$ 1-bit deltas, as in BitDelta. Red: Batched activation-product with $B$ low-rank deltas, as in S-LoRA. Left: Ablation over hidden size, assuming $N = M$ and $B = 1$. Right: Ablation over batch size, assuming $N = M = 4096$.

additional client in the batch adds an additional delta. At batch size 1 (Figure 4, right), backbone latency dominates over delta latency (BitDelta and S-LoRA) due to $W_{\text{base}}$'s $16\times$ larger memory footprint compared to a single delta. As the batch size increases (Figure 4, left), the combined memory footprint of multiple deltas exceeds $W_{\text{base}}$ around $B = 6$ to $B = 8$.

BitDelta underperforms slightly compared to S-LoRA in large-batch settings as the LoRA kernel is highly optimized for GPU. We emphasize that closing or even surpassing the gap is tractable. For example, Ma et al. [35] point out that $W_{INT1}A_{FP16}$ requires no multiplication operations and that new hardware can be co-designed with this in mind to drastically reduce energy/latency costs.

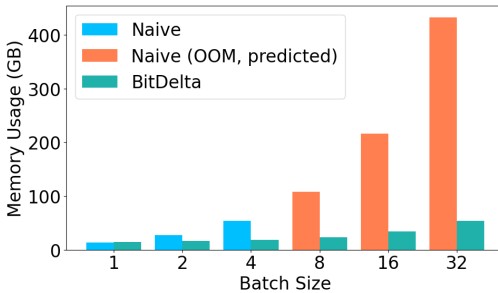 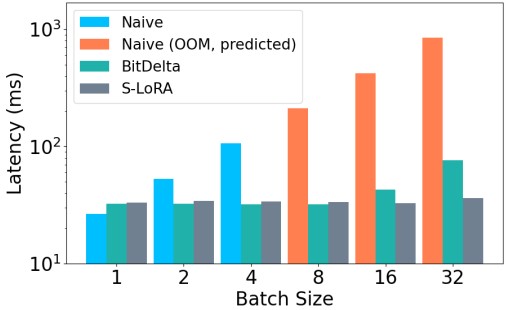

Figure 5: Memory usage of *Llama 2-7B*, assuming each sequence in the batch has a length of 128. Blue: Memory usage of the naive method, separately storing $B$ distinct fine-tuned models. Orange: Projected values for the naive method. Green: Memory usage of BitDelta. The naive forward pass succumbs to GPU memory issues at higher batch sizes.

Figure 6: End-to-end decoding latency of *Llama 2-7B*. Blue: Naive forward pass with $B$ distinct fine-tuned models. Orange: Projected values for the naive forward pass. Green: Batched forward pass with BitDelta. Gray: Batched forward pass with S-LoRA. The naive forward pass succumbs to GPU memory issues at higher batch sizes.

**End-to-end latency.** We benchmark the end-to-end decoding latency on *Llama 2-7B* variants with an input length of 128 (we find the decoding latency is less sensitive to the input length), ablated across batch size. For BitDelta and S-LoRA, the forward pass consists of the addition of two components: a single backbone pass (batch independent) and a delta pass (scales with batch size).

We compare BitDelta and S-LoRA with a naive method that computes each $W_iX_i$ separately in the forward pass. This naive approach scales poorly with batch size as it effectively maintains a separate backbone ($W_i$) for each client in the batch. Given the substantial memory footprint of the backbone, this leads to significant memory usage as batch size increases. In contrast, BitDelta and S-LoRA share a single backbone across all clients in the batch, with only the $16\times$ smaller deltas scaling with batch size. This allows for more efficient memory utilization and better performance at larger batch sizes.

We find that BitDelta and S-LoRA introduce overhead when the batch size is low. However, BitDelta and S-LoRA scale better and successfully translate the saved GPU memory to improved decoding latency, starting at $B = 2$. This is exacerbated at larger batch sizes, where the naive approach succumbs to out-of-memory issues and BitDelta and S-LoRA are still performant. In the $B \geq 16$ regime, used in modern serving solutions, BitDelta has a $>10\times$ lower per-user decoding latency than the naive method.

## 5    Conclusion

We propose BitDelta, a simple but effective approach to efficiently quantifyings the weight delta arising from the fine-tuning of LLMs down to 1 bit. BitDelta encodes the sign bits of the weight delta and a per-weight matrix scaling factor, which is calibrated further through distillation. This allows for representing multiple full-parameter fine-tuned models with one base model and multiple 1-bit deltas, enhancing applications in multi-tenancy serving by reducing GPU memory requirements and improving generation latency. BitDelta is fast and accurate, showcasing minimal performance degradation, and opens new avenues for efficient model deployment and resource utilization in machine learning.

## Acknowledgments and Disclosure of Funding

We thank Together AI, MyShell AI, National Science Foundation (NSF), MIT-IBM Watson AI Lab, MIT AI Hardware Program, and MIT Amazon Science Hub for supporting this research. JDL acknowledges support of NSF CCF 2002272, NSF IIS 2107304, NSF CIF 2212262, ONR Young Investigator Award, and NSF CAREER Award 214494. KL acknowledges the support from Meta, DataX grant from Princeton University's Center for Statistics and Machine Learning, and innovation grant from Princeton's School of Engineering and Applied Science.

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

# A Appendix

## A.1 Societal Impact

**Democratization of Fine-tuned Models.** By dramatically reducing the hardware requirements for serving fine-tuned models, BitDelta enables smaller entities to deploy state-of-the-art models more feasibly. This can accelerate innovation and application development across various industries and academic fields, making fine-tuned models accessible to a wider audience.

**Dealignment Mitigation.** BitDelta is a lossy compression method on the fine-tune information in LLMs. As such, crucial alignment information may be lost in the process of compression. We believe this is an important consequence to highlight, as BitDelta democratizes multi-tenant applications which may exacerbate this dealignment concern. We encourage further work on evaluation techniques to detect alignment loss in BitDelta, which can lead to the creation of robust methods for its mitigation.

## A.2 Additional Experiments

Table 7: We train a $r = 16$ LoRA finetune of *Llama 2-7B* on 1 epoch of UltraChat [17] and apply BitDelta with minimal performance degradation. This further shows the generality of BitDelta, which works on parameter-efficient fine-tunes in addition to full-parameter fine-tunes.

| Model/Method | ARC | BBH | HellaSwag | TruthfulQA | LAMBADA | WinoGrande | GSM8K | Average ↑ | MT-Bench |
|---|---|---|---|---|---|---|---|---|---|
| *Llama 2-7B* | 52.56 | 33.76 | 78.96 | 38.96 | 68.39 | 68.98 | 13.57 | 50.74 | – |
| *Llama 2-7B UltraChat* | 54.52 | 34.14 | 78.99 | 46.84 | 70.83 | 69.53 | 14.71 | 52.79 | 4.93 |
| BitDelta | 54.61 | 34.28 | 79.10 | 46.60 | 70.58 | 69.30 | 15.16 | 52.80 | 4.87 |

Table 8: Full results of the application of BitDelta to quantized base models, corresponding to Table 6.

| Base Model | Method | ARC | BBH | HellaSwag | TruthfulQA | LAMBADA | WinoGrande | GSM8K | Average ↑ | MT-Bench |
|---|---|---|---|---|---|---|---|---|---|---|
| | FP16 | 53.58 | 33.84 | 78.58 | 45.32 | 66.58 | 66.46 | 22.74 | 52.44 | 6.56 |
| Baseline | LLM.int8() | 53.24 | 33.71 | 78.62 | 45.02 | 66.5 | 66.06 | 22.29 | 52.21 | 6.28 |
| | GPTQ | 51.88 | 33.54 | 77.17 | 44.92 | 65.32 | 65.43 | 19.48 | 51.11 | 5.90 |
| | FP16 + $\Delta$ | 54.44 | 33.85 | 78.31 | 44.95 | 66.66 | 66.14 | 20.24 | 52.08 | 6.47 |
| *Llama 2-7B* | LLM.int8() + $\Delta$ | 53.67 | 33.48 | 78.57 | 44.71 | 66.7 | 66.85 | 19.86 | 51.98 | 6.16 |
| | GPTQ + $\Delta$ | 51.45 | 33.90 | 78.06 | 42.52 | 66.85 | 65.82 | 19.94 | 51.22 | 6.02 |
| *Llama 2-7B Chat* | GPTQ + $\Delta$ | 52.56 | 33.65 | 77.54 | 44.63 | 65.81 | 66.30 | 22.14 | 51.80 | 6.11 |

Table 9: Full results of the ablation over the fidelity of $\Delta$, corresponding to Figure 3.

| # bits in $\Delta$ | ARC | BBH | HellaSwag | TruthfulQA | LAMBADA | WinoGrande | GSM8K | Average ↑ |
|---|---|---|---|---|---|---|---|---|
| *Llama 2-7b* | 52.56 | 33.76 | 78.96 | 38.96 | 68.39 | 68.98 | 13.57 | 50.74 |
| 1 bit | 54.27 | 36.57 | 77.90 | 49.97 | 65.20 | 69.46 | 20.17 | 53.36 |
| 2 bits | 54.44 | 36.78 | 77.71 | 49.69 | 65.26 | 69.22 | 20.62 | 53.39 |
| 3 bits | 54.27 | 36.94 | 77.58 | 49.90 | 65.11 | 70.09 | 19.48 | 53.34 |
| 4 bits | 54.18 | 36.94 | 77.54 | 49.80 | 64.95 | 69.53 | 19.18 | 53.16 |
| 5 bits | 53.67 | 36.78 | 77.63 | 50.15 | 65.22 | 69.69 | 18.57 | 53.10 |
| 6 bits | 53.67 | 36.85 | 77.64 | 50.20 | 65.07 | 69.69 | 18.80 | 53.13 |
| 7 bits | 53.74 | 37.01 | 77.56 | 50.29 | 65.15 | 69.38 | 18.50 | 53.09 |
| 8 bits | 53.84 | 36.94 | 77.51 | 50.15 | 64.95 | 70.17 | 18.80 | 53.19 |
| *Vicuna-7b v1.5* | 53.92 | 37.14 | 77.45 | 50.36 | 64.41 | 69.61 | 19.03 | 53.13 |

Table 10: Full results of BitDelta applied to fine-tuned models in the Llama-2 and Mistral families, corresponding to Table 2.

| Model | Method | ARC | BBH | HellaSwag | TruthfulQA | LAMBADA | WinoGrande | GSM8K | **Average ↑** | MT-Bench ↑ |
|---|---|---|---|---|---|---|---|---|---|---|
| *Llama 2-7B* | – | 52.56 | 33.76 | 78.96 | 38.96 | 68.39 | 68.98 | 13.57 | 50.74 | – |
| *Llama 2-7B Chat* | Baseline | 53.58 | 33.84 | 78.58 | 45.32 | 66.58 | 66.46 | 22.74 | 52.44 | 6.56 |
| | BitDelta-Initial | 55.46 | 35.56 | 76.32 | 41.10 | 68.14 | 68.03 | 18.27 | 51.84 | 6.31 |
| | BitDelta | 54.44 | 33.85 | 78.31 | 44.95 | 66.66 | 66.14 | 20.24 | 52.08 | 6.47 |
| *Vicuna-7B v1.5* | Baseline | 53.92 | 37.14 | 77.45 | 50.36 | 64.41 | 69.61 | 19.03 | 53.13 | 6.04 |
| | BitDelta-Initial | 54.69 | 36.74 | 78.47 | 47.63 | 66.31 | 68.75 | 19.56 | 53.16 | 5.67 |
| | BitDelta | 54.27 | 36.57 | 77.9 | 49.97 | 65.2 | 69.46 | 20.17 | 53.36 | 5.99 |
| *Vicuna-7B v1.5 16k* | Baseline | 54.86 | 35.63 | 77.06 | 50.38 | 52.32 | 67.64 | 14.18 | 50.30 | 6.06 |
| | BitDelta-Initial | 55.55 | 33.24 | 77.99 | 45.58 | 56.8 | 68.98 | 13.95 | 50.30 | 5.69 |
| | BitDelta | 54.61 | 34.68 | 77.14 | 48.75 | 53.89 | 67.88 | 14.48 | 50.20 | 6.24 |
| *Xwin LM-7B v0.1* | Baseline | 57.59 | 34.05 | 79.15 | 48.06 | 68.02 | 69.22 | 10.77 | 52.41 | 6.24 |
| | BitDelta-Initial | 56.40 | 33.90 | 80.26 | 44.56 | 69.86 | 69.14 | 16.68 | 52.97 | 5.79 |
| | BitDelta | 57.94 | 34.19 | 79.36 | 47.62 | 68.29 | 69.53 | 9.02 | 52.28 | 6.50 |
| *Llama 2-13B* | – | 59.47 | 39.03 | 82.23 | 36.90 | 70.44 | 72.22 | 22.74 | 54.72 | – |
| *Llama 2-13B Chat* | Baseline | 60.32 | 37.89 | 82.15 | 43.95 | 68.62 | 70.96 | 33.13 | 56.72 | 6.98 |
| | BitDelta-Initial | 59.90 | 38.04 | 82.13 | 41.70 | 69.82 | 71.35 | 33.36 | 56.61 | 7.06 |
| | BitDelta | 59.98 | 38.03 | 81.92 | 43.47 | 68.46 | 71.43 | 31.92 | 56.46 | 6.95 |
| *Vicuna-13B v1.5* | Baseline | 57.34 | 39.47 | 81.14 | 50.86 | 68.48 | 71.67 | 29.72 | 56.95 | 6.48 |
| | BitDelta-Initial | 54.69 | 36.74 | 78.47 | 47.63 | 66.31 | 68.75 | 31.84 | 54.92 | 6.51 |
| | BitDelta | 57.42 | 39.20 | 81.33 | 50.39 | 68.81 | 71.51 | 30.48 | 57.02 | 6.81 |
| *Vicuna-13B v1.5 16k* | Baseline | 54.86 | 35.63 | 77.06 | 50.38 | 52.32 | 67.64 | 29.72 | 52.52 | 6.90 |
| | BitDelta-Initial | 59.90 | 38.04 | 82.13 | 41.70 | 69.82 | 71.35 | 26.76 | 55.67 | 6.60 |
| | BitDelta | 54.61 | 34.68 | 77.14 | 48.75 | 53.89 | 67.88 | 28.73 | 52.24 | 6.88 |
| *WizardLM-13B v1.2* | Baseline | 60.15 | 40.82 | 82.58 | 47.17 | 69.26 | 71.90 | 42.38 | 59.18 | 6.95 |
| | BitDelta-Initial | 60.41 | 40.27 | 83.26 | 44.89 | 70.23 | 71.74 | 42.08 | 58.98 | 6.73 |
| | BitDelta | 60.92 | 41.30 | 82.55 | 46.67 | 68.97 | 71.51 | 41.62 | 59.08 | 6.93 |
| *Xwin LM-13B v0.1* | Baseline | 63.14 | 40.12 | 82.92 | 45.54 | 70.62 | 73.09 | 21.15 | 56.65 | 6.78 |
| | BitDelta-Initial | 63.4 | 40.33 | 83.71 | 43.6 | 71.26 | 73.09 | 26.76 | 57.45 | 6.70 |
| | BitDelta | 62.80 | 39.81 | 83.01 | 48.19 | 70.74 | 72.30 | 21.76 | 56.94 | 6.83 |
| *Llama 2-70B* | – | 67.58 | 51.67 | 87.00 | 44.82 | 74.81 | 77.98 | 52.69 | 65.22 | – |
| *Llama 2-70B Chat* | Baseline | 65.44 | 43.93 | 85.91 | 52.77 | 73.90 | 74.90 | 47.61 | 63.49 | 7.12 |
| | BitDelta-Initial | 63.4 | 38.67 | 81.36 | 41.63 | 72.66 | 73.95 | 42.38 | 59.15 | 6.85 |
| | BitDelta | 65.87 | 44.97 | 85.65 | 51.37 | 74.29 | 74.90 | 48.82 | 63.70 | 7.06 |
| *Solar-0-70B* | Baseline | 71.16 | 55.54 | 87.78 | 62.03 | 75.04 | 79.32 | 56.18 | 69.58 | 7.07 |
| | BitDelta-Initial | 69.54 | 54.52 | 87.57 | 59.08 | 75.37 | 78.69 | 56.79 | 68.79 | 6.79 |
| | BitDelta | 70.82 | 55.06 | 87.35 | 62.03 | 75.86 | 78.77 | 56.63 | 69.50 | 6.82 |
| *Xwin LM-70B v0.1* | Baseline | 70.65 | 52.40 | 87.15 | 60.06 | 75.04 | 78.06 | 40.33 | 66.24 | 7.45 |
| | BitDelta-Initial | 69.97 | 52.93 | 87.36 | 60.77 | 75.51 | 78.14 | 50.64 | 67.90 | 7.70 |
| | BitDelta | 70.22 | 52.22 | 86.97 | 58.57 | 75.49 | 77.58 | 40.18 | 65.89 | 7.34 |
| *Mistral-7B v0.1* | – | 61.35 | 41.18 | 83.46 | 42.60 | 70.10 | 73.80 | 37.76 | 58.61 | – |
| *Mistral-7B v0.1 Instruct* | Baseline | 55.03 | 38.66 | 75.52 | 55.93 | 63.28 | 69.30 | 32.75 | 55.78 | 6.86 |
| | BitDelta-Initial | 59.22 | 40.25 | 79.91 | 51.27 | 67.63 | 72.14 | 38.82 | 58.46 | 6.54 |
| | BitDelta | 55.38 | 37.95 | 75.62 | 55.23 | 66.06 | 70.48 | 31.54 | 56.04 | 6.43 |
| *Zephyr-7B-β* | Baseline | 63.82 | 39.04 | 84.33 | 55.12 | 66.23 | 72.69 | 34.34 | 59.37 | 7.18 |
| | BitDelta-Initial | 63.57 | 41.87 | 83.85 | 54.53 | 67.73 | 73.56 | 40.26 | 60.77 | 6.70 |
| | BitDelta | 65.02 | 41.64 | 84.05 | 58.39 | 66.33 | 73.95 | 31.92 | 60.19 | 7.00 |
| *OpenChat 3.5* | Baseline | 64.51 | 45.28 | 84.39 | 47.34 | 65.19 | 72.61 | 68.84 | 64.02 | 7.74 |
| | BitDelta-Initial | 64.16 | 45.23 | 84.13 | 43.34 | 68.62 | 77.43 | 57.77 | 62.95 | 5.71 |
| | BitDelta | 64.93 | 44.57 | 84.44 | 46.24 | 65.88 | 76.40 | 57.70 | 62.88 | 7.38 |
| *Dolphin 2.2.1* | Baseline | 64.16 | 44.49 | 83.30 | 54.02 | 69.36 | 75.22 | 54.28 | 63.55 | 7.36 |
| | BitDelta-Initial | 64.16 | 44.43 | 84.01 | 48.14 | 69.98 | 75.30 | 50.27 | 62.33 | 7.10 |
| | BitDelta | 64.59 | 43.08 | 83.44 | 54.91 | 68.39 | 75.37 | 52.84 | 63.23 | 7.20 |
| *OpenOrca-7B* | Baseline | 62.80 | 44.45 | 83.58 | 52.30 | 66.10 | 73.24 | 50.11 | 61.80 | 6.70 |
| | BitDelta-Initial | 63.74 | 44.46 | 84.15 | 49.66 | 69.05 | 74.03 | 49.96 | 62.15 | 7.12 |
| | BitDelta | 63.65 | 43.46 | 83.49 | 51.67 | 66.12 | 74.27 | 49.58 | 61.75 | 7.05 |

