# OpenReview forum: "BitDelta: Your Fine-Tune May Only Be Worth One Bit"
_NeurIPS.cc/2024/Conference — NeurIPS 2024 poster_

### Official Review · Reviewer_b1KL · 2024-06-25

**Soundness:** 2
**Presentation:** 3
**Contribution:** 2
**Rating:** 4
**Confidence:** 3

**Summary:**

This paper introduces BitDelta which quantizes the aggregated weight updates (the authors call it “delta”) to 1-bit after full fine-tuning. The paper claims that the approach has two applications: 1. First, it shows that the delta is highly redundant and 2. It is useful in the multi-client-single-server applications where the high precision model will be saved on the server and each client only store it’s own 1-bit delta. They show that in this scenario, the generation takes up to 10x memory reduction (and similar in latency improvement).

**Strengths:**

1. The paper studies an important problem for fine-tuning LLMs with a new approach where they quantizes the delta to 1-bit,
2. The experiments are done on the most important models like LLaMa-2 and Mistral,
3. The paper provides a kernel for INT1xFP16 matrix multiplication.

**Weaknesses:**

1, The author claim that BitDelta shows the potential redundancy of information added during fine-tuning. However, this is not a new finding and almost all PEFT approaches (for example LoRA) are based on this fact.

2. It seems that the paper completely missed the full fine-tuning costs and just measured the memory/latency of the serving step. However, I would suggest to have “apple to apple” comparisons and compare the fine-tuning+serving cost (including memory and runtime) of “full-fine tuning+1bit optimization+serving” against “fine-tuning+serving” in PEFT schemes (like LoRA).

3. In Table 6, the paper shows higher accuracy for FP+delta compared to GPTQ. Again, I would rather like to see memory Vs. accuracy tradeoff in such comparisons as a function of number of clients. The fact is FP+delta does not have the same memory as GPTQ (please current me if I am wrong).

4. It would be nice to have a performance model for the latency of the decoding as a function of client numbers. This is also missed and needs to be include as this is the main claim of the paper.

**Questions:**

Please check the "weaknesses" section. I would be happy to discuss and change my score.

**Limitations:**

I think the authors should define and present the limitations of the method more clearly.

---

> ### Author Rebuttal · Authors · 2024-08-02
>
> We thank the reviewer for the kind review! Please find below our point-by-point response regarding your feedback:
>
> > 1, The author claim that BitDelta shows the potential redundancy of information added during fine-tuning. However, this is not a new finding and almost all PEFT approaches (for example LoRA) are based on this fact.
>
> We agree that the redundant fine-tune information angle is not new, hence our analogy that LoRA enforces structure **during training**. However, we believe there is nontrivial novelty in using this idea to accurately quantize the weight delta **post training to 1-bit** for full-parameter fine-tuned models, and successfully translating this reduced memory consumption to a >10x wall clock speedup in multi-tenant settings.
>
> > 2. It seems that the paper completely missed the full fine-tuning costs and just measured the memory/latency of the serving step. However, I would suggest to have “apple to apple” comparisons and compare the fine-tuning+serving cost (including memory and runtime) of “full-fine tuning+1bit optimization+serving” against “fine-tuning+serving” in PEFT schemes (like LoRA).
>
> We respectfully disagree with the statement's premise -- to clarify, we do not fine-tune our own models, and instead target existing popular SOTA fine-tuned models on platforms like HuggingFace, as this setting is where our methodology’s downstream applications are most relevant. As such, the effective cost of such models is amortized over many people around the world. If we were in a different setting and needed to fine-tune our own models from scratch (eg. in a niche domain), then we would agree that a full apples-to-apples comparison including the cost of fine-tuning would be more appropriate.
>
> However, the reviewer raises an interesting point and we plan to clarify this in the final manuscript to ensure the value proposition of our work is more accurately understood.
>
> > 3. In Table 6, the paper shows higher accuracy for FP+delta compared to GPTQ. Again, I would rather like to see memory Vs. accuracy tradeoff in such comparisons as a function of number of clients. The fact is FP+delta does not have the same memory as GPTQ (please current me if I am wrong).
>
> The reviewer is correct in that $FP16+\Delta$ has a different memory footprint than $GPTQ$. The crossover point would be about 5 models. Serving separate quantized models is mainly relevant in low-batch (low number of clients) and low-memory settings. However, as shown in Table 6, BitDelta can also be applied to quantized base models, which is a viable solution in such settings. For example, when serving 3 models, it is preferable to represent them as one 8-bit base model plus three 1-bit deltas, instead of three separate 4-bit models, in terms of both accuracy and memory.
>
> Nonetheless, BitDelta is not intended to be useful in this regime (low-batch + low-memory), and the quantization ablation mainly serves to show the robustness of the method in terms of accuracy. However, the remark that $FP16+\Delta$ outperforms $GPTQ$ may mislead readers to overgeneralize, which we will address in the final manuscript.
>
> > 4. It would be nice to have a performance model for the latency of the decoding as a function of client numbers. This is also missed and needs to be include as this is the main claim of the paper.
>
> The End-to-End Latency section (Figure 5) addresses decoding speed as a function of batch size (number of clients). We're more than happy to provide additional results if this is not what the reviewer is expecting.

---

> > ### Comment · Reviewer_b1KL · 2024-08-11
> > **Reply**
> >
> > Thanks for your answers. The authors claim that they do not present a fine-tuning scheme, but this rather a "serving" approach. I think they should highlight this as the main message of the paper and re-defining the evaluation approach. I would stick to my current score.

---

> ### Author Response · Authors · 2024-08-12
> **Followup**
>
> We thank the reviewer for the response. We will make sure to properly position the paper in the final manuscript. Though, we are wondering if the reviewer could clarify how they think our evaluation approach should be re-defined. Our baselines (For accuracy, fine-tuned models without BitDelta applied. For latency, serving fine-tuned models separately.) are fairly reasonable in this context. Are there specific aspects that the reviewer thinks are misaligned?
>
> Given that we have additionally addressed the other concerns (fine-tuning costs, latency results, etc.), we would greatly appreciate it if the reviewer could reconsider their score.

---

### Official Review · Reviewer_dvPC · 2024-07-08

**Soundness:** 3
**Presentation:** 3
**Contribution:** 2
**Rating:** 5
**Confidence:** 5

**Summary:**

Aiming at storage and serving overhead caused by multiple finetuned LLMs for various downstream tasks, this paper proposes a memory-friendly model compression method namely BitDelta which binaries the delta of each weight matrix and uses self-distillation to learn optimal scaling factors.

**Strengths:**

BitDelta innovatively decomposes a fine-tuned model into its pretrained version and an additional weight matrix delta and then successfully binary delta to reduce memory overheads while preventing large performance degradation.

**Weaknesses:**

The paper lacks innovation and has several points that do not hold up under scrutiny.

For the first contribution, the paper proposes decomposing multiple fine-tuned models into a shared pretrained model and their respective deltas, and then binarizing the deltas. The specific decomposition method is not described in the paper. Based on experience, this can be understood as using LoRA to fine-tune LLMs and binarizing the learned low-rank matrices. This is not novel, as there are already existing low-bit model fine-tuning methods, such as Q-LoRA and QA-LoRA, which are more memory-friendly and do not require retaining an additional fp16 model.

Regarding the second contribution, which involves using self-distillation to learn scaling factors, the paper's description is insufficient. For example, the initialization method of these factors is not well-explained. Furthermore, if the entire fp model's output is used to supervise the quantized model, the computational resource consumption is high. The paper could explore layer-wise optimization mechanisms to address this issue.

**Questions:**

1) The author should describe the advantages of BitDelta compared to PEFT methods, such as QLoRA, QA-LoRA, and LoftQ. Because they don't require saving the fp16 pretrained model, which results in more efficient storage utilization.
2) More detailed description should be given, such as GPTQ+Δ in Table 6. Additionally, the performance of FP16+Δ being superior to 4-bit GPTQ and 2-bit Quip does not entirely prove that directly quantizing the base model is impractical, as INT8-RTN still shows better performance. If we consider 8-bit GPTQ, or other sota PTQ methods such as Omniquant, AWQ, it might also perform better while reducing memory usage by half. Therefore, it is necessary to provide additional arguments from other perspectives to explain why directly quantizing the base model is not preferable.
Extensive additional experiments are not necessary, but it is important to clearly explain the aforementioned issues.

**Limitations:**

The authors have addressed the limitations.

---

> ### Author Rebuttal · Authors · 2024-08-02
>
> We thank the reviewer for the kind review! Please find below our point-by-point response regarding your feedback:
>
> We would first like to clarify a misconception that significantly impacts the evaluation of our work: **we do not fine-tune our own models** in this paper. Rather, we take existing popular SOTA full-parameter fine-tuned models, and compress the weight delta between it and its underlying base model. This is done in a hardware friendly way, such that inference on the deltas is fast and performant.
>
> Fundamentally, the goal of BitDelta is to unlock the efficient multi-tenant serving of **existing SOTA full-parameter fine-tuned models** on platforms like HuggingFace. Methods like QLoRA are impactful in that they democratize fine-tuning in resource constrained settings, trading off accuracy for decreased memory footprint. This is most useful in settings like fine-tuning your own models locally (eg. when you’re in a niche domain) and on the edge. Because of the differing target use cases, it’s hard to make a useful comparison that adequately captures the value propositions of both methods.
>
> W1:
> > The specific decomposition method is not described in the paper.
>
> We describe the decomposition method in Section 3.1, lines 117-126. The base model and fine-tuned model weights are known apriori, and the decomposition is an element-wise subtraction.
>
> > Based on experience, this can be understood as using LoRA to fine-tune LLMs and binarizing the learned low-rank matrices. This is not novel, as there are already existing low-bit model fine-tuning methods, such as Q-LoRA and QA-LoRA, which are more memory-friendly and do not require retaining an additional fp16 model.
>
> We do not fine-tune the LLMs ourselves, please see our beginning statement.
>
> W2:
> > Regarding the second contribution, which involves using self-distillation to learn scaling factors, the paper's description is insufficient. For example, the initialization method of these factors is not well-explained.
>
> We apologize for the unclear presentation and will revise lines 122-126. The initialization of the scaling factors is set to the mean of the absolute values of the per-tensor weight entries. This is done to minimize weight quantization error with respect to L2 norm.
>
> >  Furthermore, if the entire fp model's output is used to supervise the quantized model, the computational resource consumption is high. The paper could explore layer-wise optimization mechanisms to address this issue.
>
> In Section 3.2 we describe the methodology cost, and conclude that the total computational cost is fairly similar to other PTQ methods. To be clear, the methodology assumes the existence of a trained base model, and a trained fine-tuned model, and does not fine-tune in the sense that Q-LoRA + its variants do.
>
> One potential optimization not mentioned would be to precompute the teacher model logits (which should not be a storage issue given the low sample lengths + number of samples), which would halve the memory footprint. Nonetheless, we agree that it may be possible to further optimize this. For example, we could search for the optimal per tensor scaling factor through a grid search, loading one tensor at a time, similar to AWQ [1].
>
> > Q1: The author should describe the advantages of BitDelta compared to PEFT methods, such as QLoRA, QA-LoRA, and LoftQ. Because they don't require saving the fp16 pretrained model, which results in more efficient storage utilization.
>
> We do not fine-tune the LLMs ourselves, please see our beginning remark. Additionally, as shown in Table 6, BitDelta also works well in conjunction with quantized base models.
>
> > Q2: More detailed description should be given, such as GPTQ+Δ in Table 6. Additionally, the performance of FP16+Δ being superior to 4-bit GPTQ and 2-bit Quip does not entirely prove that directly quantizing the base model is impractical, as INT8-RTN still shows better performance. If we consider 8-bit GPTQ, or other sota PTQ methods such as Omniquant, AWQ, it might also perform better while reducing memory usage by half. Therefore, it is necessary to provide additional arguments from other perspectives to explain why directly quantizing the base model is not preferable. Extensive additional experiments are not necessary, but it is important to clearly explain the aforementioned issues.
>
> We did not intend to suggest that directly quantizing the base model is not preferable, and we apologize for any confusion. Our goal was to demonstrate the orthogonality of BitDelta to quantizing the base model. Given a more stringent memory constraint, providers are able to quantize the base model in addition to applying BitDelta, without significant degradation in performance. In fact, applying 8-bit quantization might have such a negligible impact on accuracy that it could be preferable purely from an inference speed perspective, regardless of memory constraints.
>
> Our corollary statement ($FP16+\Delta$ outperforms $GPTQ$) was an interesting observation we noticed. However, we acknowledge this may not necessarily be true for stronger baselines (AWQ, Omniquant, etc.). We will moderate our claims in the final manuscript to better reflect the
> scope of this finding.
>
> [1]: AWQ: Activation-aware Weight Quantization for LLM Compression and Acceleration: https://arxiv.org/abs/2306.00978

---

### Official Review · Reviewer_dXr8 · 2024-07-11

**Soundness:** 2
**Presentation:** 4
**Contribution:** 3
**Rating:** 6
**Confidence:** 4

**Summary:**

This paper introduces Bitdelta, a method that enables quantization with just 1 bit. The main idea is to compress the weight delta into a scalar and a binary matrix. The experimental results demonstrate that Bitdelta achieves better performance compared to other techniques.

**Strengths:**

- Easy to read and well-structured.
- The authors effectively demonstrate that their proposed technique, Bitdelta, performs well compared to existing quantization techniques.
- It is expected that Bitdelta can be extended to areas such as multi-tenant serving systems.

**Weaknesses:**

- A significant contribution of Bitdelta lies in its ability to reduce memory consumption through 1-bit quantization. However, the paper lacks detailed evidence to support this claim. It is necessary to compare memory consumption for each compression technique, but currently, only Bitdelta is shown.
- It would be better for the paper to mention and explain the figures within the proper location of text. For example, Figure 1 and Figure 4 are included but not mentioned in the text. It is hard to understand without sufficient explanation.
- Table 1 lacks information about the base models for Bitdelta and SVD. It should clearly specify whether these are based on Llama-7B or Llama-7B Chat.

**Questions:**

- I am curious about the clear differences between Bitdelta and other compression technologies. For example, can Bitdelta, which adopts the Post-Training Quantization (PTQ) method, be used in combination with other PTQ techniques? Also, what happens if you combine Bitdelta with compression techniques like pruning? It would be beneficial to discuss the relationships between various compression technologies
- (lines 165-167) The authors said that generation latency is proportional to the GPU memory used. How can this claim be proven? It would be helpful to mention references or provide supporting data
- I think BitNet has a similar purpose with the design of 1-bit quantization. Comparing Bitdelta to BitNet is required

**Limitations:**

I think the paper will be strengthened with the measurements on GPU (or memory) consumption are added

---

> ### Author Rebuttal · Authors · 2024-08-02
>
> We thank the reviewer for the kind review! Please find below our point-by-point response regarding your feedback:
>
> > A significant contribution of Bitdelta lies in its ability to reduce memory consumption through 1-bit quantization. However, the paper lacks detailed evidence to support this claim. It is necessary to compare memory consumption for each compression technique, but currently, only Bitdelta is shown.
>
> We would appreciate further clarification on this point -- is the reviewer referencing compression techniques like quantization (GPTQ,AWQ), or potentially the SVD baseline in Table 1? Nevertheless, we are considering delta compression in this work, which differs from classic quantization methods.
>
> > It would be better for the paper to mention and explain the figures within the proper location of text. For example, Figure 1 and Figure 4 are included but not mentioned in the text. It is hard to understand without sufficient explanation.
>
> > Table 1 lacks information about the base models for Bitdelta and SVD. It should clearly specify whether these are based on Llama-7B or Llama-7B Chat.
>
> We apologize for the unclear presentation and thank the reviewer for the suggestion. We will fix these issues in the final manuscript. The Table 1 results are based on compressing the weight delta between Llama-7B and Llama-7B Chat, so the resultant model is an approximation of Llama-7B Chat.
>
> > I am curious about the clear differences between Bitdelta and other compression technologies. For example, can Bitdelta, which adopts the Post-Training Quantization (PTQ) method, be used in combination with other PTQ techniques? Also, what happens if you combine Bitdelta with compression techniques like pruning? It would be beneficial to discuss the relationships between various compression technologies
>
> In Table 6 we have results where we apply PTQ in conjunction with BitDelta -- we found that the two methods are fairly orthogonal. We expect other methods (pruning, etc.) that also apply to the base model to also be fairly orthogonal.
>
> Regarding further compression of the delta, it may be possible to employ techniques such as vector quantization and incoherence processing (similar to QuIP# [1]) to achieve accurate sub 1-bit deltas. However, we have to be cognizant of the hardware friendliness of these methods, and whether the reduced delta size outweighs the associated kernel overhead.
>
> > (lines 165-167) The authors said that generation latency is proportional to the GPU memory used. How can this claim be proven? It would be helpful to mention references or provide supporting data
>
> The memory bound nature of LLM decoding is well documented [2]. During the decoding phase (on modern GPUs), the time taken to transfer weights and KV caches to GPU registers far outweighs the time needed to compute the associated skinny matrix multiplications.
>
> AWQ [3] leverages this to translate a reduction in memory footprint (through weight quantization) to a ~3x wall clock speedup. We likewise translate a reduction in memory footprint (through representing multiple fine-tuned weights with just one base weight and multiple compressed deltas) to a ~10x wall clock speedup when concurrently serving 16 models.
>
> > I think BitNet has a similar purpose with the design of 1-bit quantization. Comparing Bitdelta to BitNet is required
>
> BitNet fundamentally has a different purpose in that they propose a new architecture based on 1-bit weight entries for LLM **pretraining**, with the goal of showing superiority over conventional 16-bit pretraining. BitDelta differs in that it compresses the weight delta of two **existing pretrained** 16-bit models to 1-bit, while keeping the base model in 16-bit precision, with the goal of unlocking efficient multi-tenant serving of full-parameter fine-tuned models. The two are related only in that they both use $W_\text{INT1}$ matrix operations.
>
> [1]: QuIP#: Even Better LLM Quantization with Hadamard Incoherence and Lattice Codebooks: https://arxiv.org/abs/2402.04396
>
> [2]: LLM Inference Unveiled: Survey and Roofline Model Insights: https://arxiv.org/abs/2402.16363
>
> [3]: AWQ: Activation-aware Weight Quantization for LLM Compression and Acceleration: https://arxiv.org/abs/2306.00978

---

> > ### Comment · Reviewer_dXr8 · 2024-08-11
> >
> > - Thank you for your efforts in addressing the weaknesses and questions. Based on the authors’ responses, I understand where I was confused, and I have updated my review decision (borderline accept -> weak accept).
> > - Also, although the authors logically explained the memory consumption aspects, the actual memory consumption of the technique is affected by various factors; therefore, providing the actual numbers would be beneficial.

---

### Official Review · Reviewer_dEGq · 2024-07-12

**Soundness:** 3
**Presentation:** 4
**Contribution:** 3
**Rating:** 7
**Confidence:** 4

**Summary:**

The paper proposes to quantize the weight delta of a fine-tuned LLM to 1-bit and observes that the model quality only drops a little.
During the binarization step, it requires calibrating the scaling factor with a few hundreds of distillation steps. This is less than a full fine tuning. Evaluation show that the proposed method produces higher model quality than other post-training quantization techniques such as GPTQ and QuIP.

**Strengths:**

The paper discovered a nice trade-off between fine-tuned model storage size and model quality. Trading 16x lower weight size by only storing the 1-bit delta for the limited quality drop as reported in the paper is impressive. Importantly, the binarization step has a near-post-training cost, instead of fine-tuning from the beginning. The paper did solid ablation studies on how important the scaling factor calibration is, which well described the effect of each component in the proposed method. The latency study is also appreciated.

**Weaknesses:**

While requiring low storage size, the proposed method introduces an extra binary-float matmul during inference. Although it is indeed a special kernel and can have much lower inference time than the float matmul, the overhead will become more significant when the base model is low-precision.

**Questions:**

1. Line 133 mentioned the scale distillation is robust to the choice of calibration dataset. Does it mean that it can also use synthetic data? It will be an improvement if the paper presents such an ablation.

2. Line 226: can the quantized delta be merged with the quantized base model? For example, by synchronizing the scaling factors.

**Limitations:**

The paper does not have negative societal impact as far as the reviewer can tell.

---

> ### Author Rebuttal · Authors · 2024-08-02
>
> We thank the reviewer for the kind review! Please find below our point-by-point response regarding your feedback:
> > While requiring low storage size, the proposed method introduces an extra binary-float matmul during inference. Although it is indeed a special kernel and can have much lower inference time than the float matmul, the overhead will become more significant when the base model is low-precision.
>
> The delta kernel overhead is indeed nontrivial in the regime where the base model is low-precision and $B$ (the number of served models) is small. Similar solutions (S-LoRA, etc.) suffer from the same issue and are actually slower than BitDelta when $B \leq 4$. Such multi-tenant solutions work best in higher batch settings.
>
> In terms of overall throughput, with a 16-bit base model (shown in Figure 5), BitDelta outperforms the naive method of running each model separately for all $B>1$. We expect a similar result for quantized base models, though potentially with a higher crossover point.
>
> > 1. Line 133 mentioned the scale distillation is robust to the choice of calibration dataset. Does it mean that it can also use synthetic data? It will be an improvement if the paper presents such an ablation.
>
> Intuitively this seems very doable, considering scale distillation already does well with generic internet data. We will try to include this in the final manuscript.
>
> > 2. Line 226: can the quantized delta be merged with the quantized base model? For example, by synchronizing the scaling factors.
>
> We don’t see an easy way to do this losslessly, but we’re happy to chat more about this. To us it seems difficult to combine quantized weight matrices that have different scale factors.

---

> > ### Comment · Reviewer_dEGq · 2024-08-13
> > **Thank the authors for response**
> >
> > Thank the authors for the response. I have read all replies and comments from other reviewers.

---

### Decision · Program_Chairs · 2024-09-25

**Decision:**

Accept (poster)

**Comment:**

The reviews are mostly favourable, although varying in their level of support for the paper. A couple of reviewers misunderstood initially the main idea and I urge the authors to pay special attention to this aspect when revising the paper. Having said that, after reading through the paper myself I did not see any parts where the paper would be impossible to understand so I do not think any misunderstandings like those could be a base for rejection. To make it more evident, both reviewers dXr8 and dvPC revised their scores after the rebuttal (although dXr8 did so silently), suggesting there are no fundamental flaws in that regard in the paper.

Only reviewer b1KL kept their score and mentioned the paper should make it clearer that its focus is on serving, and that evaluation needs to be redesigned - however, it is not very clear to me what the reviewer here had in mind. With regard to the positioning of the paper, as mentioned above, although there might be some room for improvement I do not think it warrants rejection. With regard to evaluation, the biggest shortcoming seems to be related to the fact that the proposed method only makes sense in a multi-tenancy setting, but strictly speaking no such case has been evaluated. While this might see as a serious shortcoming, given its relevance for the central point of the paper, it is important to note that the current results are sufficient to infer the expected performance in a multi-tenancy setting. More specifically, the criticism is related to the memory consumption (dXr8, b1KL) and latency (my own comment).

With respect to the memory, while Table 5 provides the core information needed to understand and validate benefits of the method, the fact the authors do not provide any end-to-end memory, especially while varying the number of fine-tuned models, is a notable omission and should be fixed.

With respect to latency, I find the current comparison to be very confusing. After carefully reading the entire section multiple times, I do not think there is any serious mistake there, but it is unnecessarily difficult to understand. For example: "batch size" referring to the number of models is very confusing, referring to only the int1-fp16 part of the method as BitDelta in Figure 4, but encompassing the entire W_baseX + deltaX in Figure 5 does not help, etc. (in general, it took me much longer than reasonably needed to understand why BitDelta vs. Backbone in Figure 4 exhibits the exact opposite trend to BitDelta vs. Naive in Figure 5). I would urge the authors to rewrite this section thoroughly.

Having said that, both of the aspects above seem to be matters pertaining more to the presentation of the current results rather than omission of important experiments needed to support claims made by the paper. Given that and the fact that, after the rebuttal, overall motivation or contributions do not seem to be questioned by the reviewers, who largely lean towards acceptance, I also recommend acceptance. Having said that, I do trust the authors to revise their paper accordingly to the feedback from the review process, as it seems there is many minor places that should be improved before publication.